# Enhancing Weaned Piglet Health and Performance: The Role of Autolyzed Yeast (*Saccharomyces cerevisiae*) and β-Glucans as a Blood Plasma Alternative in Diets

**DOI:** 10.3390/ani14040631

**Published:** 2024-02-16

**Authors:** Robson Sfaciotti Barducci, Anderson Aparecido Dias Santos, Leticia Graziele Pacheco, Thaila Cristina Putarov, João Fernando Albers Koch, Marco Aurélio Callegari, Cleandro Pazinato Dias, Rafael Humberto de Carvalho, Caio Abércio da Silva

**Affiliations:** 1Biorigin, Lençóis Paulista 18680-900, SP, Brazil; robson.barducci@biorigin.net (R.S.B.); anderson.santos@biorigin.net (A.A.D.S.); leticia.pacheco@biorigin.net (L.G.P.); thaila.putarov@biorigin.net (T.C.P.); joao.koch@biorigin.net (J.F.A.K.); 2Akei Animal Research, Fartura 18870-970, SP, Brazil; contato@akei.agr.br (M.A.C.); cleandro@cleandrodias.com.br (C.P.D.); rafael.carvalho@uel.br (R.H.d.C.); 3Animal Science Program, Center of Agrarian Sciences, State University of Londrina, Londrina 86057-970, PR, Brazil

**Keywords:** additives, cytokines, intestinal microbiota, swine, weaning

## Abstract

**Simple Summary:**

This study investigates the impact of substituting blood plasma with autolyzed yeast (*Saccharomyces cerevisiae*), both with and without an immunomodulator (1,3/1,6 β-glucans), in the diets of nursery-phase piglets. The research involved 240 piglets in the nursery phase, and assessed their zootechnical performance, intestinal health, immune responses, and energy metabolism under four dietary treatments. The findings revealed that diets including autolyzed yeast, with or without the immunomodulator, enhance feed conversion rates and reduce the need for medications compared to those containing only blood plasma. Additionally, this substitution positively influenced the piglets’ immune and metabolic health, as evidenced by changes in triglyceride levels and beta-hydroxybutyrate concentrations, and modulated the intestinal microbiota, indicating a beneficial role in piglet nutrition. These results suggest that autolyzed yeast, with or without immunomodulators, could be a viable alternative to blood plasma in piglet diets, offering health and performance benefits.

**Abstract:**

The objective of this study was to evaluate the inclusion of the autolyzed yeast (AY) *Saccharomyces cerevisiae* with or without an immunomodulator (1,3/1,6 β-glucans) as a total/partial substitute for blood plasma (BP) in the diet of post-weaning piglets; zootechnical performance, intestinal health and microbiota, immune responses and energy metabolism were assessed. A total of 240 castrated male and female piglets, with a mean age of 22 days and mean initial weight of 5.24 ± 0.82 kg, were randomly divided into blocks of four treatments with 12 replicates. The dietary inclusions were blood plasma (BP), autolyzed yeast (AY), autolyzed yeast + immunomodulator (AYI) and 50% BP and 50% AY (BPAY). In pre-initial phase II (29–35 days), piglets fed AY showed better feed conversion (FCR = 1.358) than the piglets in the BP (1.484), AYI (1.379) and BPAY (1.442) groups, i.e., 8.49% (0.126), 1.52% (0.021) and 4.50% (0.084), respectively (*p* = 0.0293). In the total period (21–42 days), better FCR was observed in the AYI (1.458) group, i.e., 4.64% (0.071), 1.15% (0.017) and 4.58% (0.070), than in the BP (1.529), AY (1.475) and BPAY (1.528) groups, respectively (*p* = 0.0150). In piglets fed AY (n = 3) and BPAY (n = 2), there was a reduction in the number of medications, i.e., 82.35% (−14n) and 88.23% (−15n), respectively (*p* = 0.0001), compared with that in the BP group (n = 17). In the AY group (73.83 mg/dL), AYI group (69.92 mg/dL), and BPAY group (69.58 mg/dL), piglets exhibited increases in triglyceride levels of 79.32%, 69.83%, and 69.00%, respectively, in comparison to those in the BP group, which had triglyceride levels of 41.17 mg/dL (*p* = 0.0400). The beta-hydroxybutyrate concentration in the AY group (79.96 ng/μL) was lower by 31.95%, 22.64%, and 5.89% compared to the BP group (117.50 ng/μL), AYI group (103.36 ng/μL), and BPAY group (84.67 ng/μL), respectively (*p* = 0.0072). In the AYI group, there was modulation of the microbiota, with an increase in the relative abundance of bacteria of the genera *Lactobacillus*, *Collinsella* and *Bulleidia*. AY, associated or not associated with an immunomodulator, is a potential substitute for BP in diets for piglets in the nursery phase, with positive effects on immune, metabolic, and intestinal microbial performance.

## 1. Introduction

The first few weeks after weaning still represent a great challenge for piglets because the combination of social, environmental, sanitary, food and nutritional stresses potentiate the impact of maternal segregation [1,2]. Weaning is known to increase the expression of genes that regulate proinflammatory cytokines, such as IL-1β, IL-6 and TNF-α, and inflammatory genes, such as CXCL10 and CCL19, which are associated with anatomical and functional disorders of the intestine [3,4].

Piglets born with low weight are more likely to have a greater presence of dysfunctional mitochondria and thus a reduced mitochondrial antioxidant capacity and suboptimal inflammatory response, processes resulting from their vulnerability to the multiple stressors that occur during the peri-weaning period [3]. Additionally, piglets from hyperprolific sows, especially those with lower birth weights, often demonstrate reduced consumption of colostrum and feed before weaning, adversely affecting their health and post-weaning performance [5,6,7].

This low feed intake induces changes in the permeability of the gastrointestinal tract, especially in the first two weeks post-weaning, with a reduction in villus height and in the depth of the crypts of the intestinal mucosa, decreasing the absorption capacity [8,9]. This heightened intestinal vulnerability coincides with a high incidence of infectious diseases, including viral infections and pathogenic bacteria such as *E. coli*, often leading to typical post-weaning diarrhea. This elevates the frequency of diarrhea, thereby interfering with performance and leading to more frequent instances of dehydration and sudden death [10,11,12].

The minimization of the effects of post-weaning stress while preserving the intestinal health status involves, in addition to management actions, the use of a set of feed additives that include peptides, organic and inorganic acids, prebiotics, probiotics, plant extracts, essential oils, and yeasts, among others [13], and the formulation of diets with high-quality ingredients, which are aimed at stimulating consumption, improving digestibility, and promoting the immunological profile.

In this context, blood plasma (BP), which is obtained from the industrial fractionation of healthy pig or cattle blood, stands out as one of the noblest sources of protein in animal feed, containing high levels of vitamins and minerals such as vitamin B12, calcium, phosphorus, sodium, chloride, potassium, and magnesium [14]. In addition, BP is an excellent source of essential amino acids, and is commonly superior to other protein sources, such as dairy products [15,16]. In addition to being a rich source of highly digestible proteins, blood plasma (BP) contains elevated levels of immunoglobulins (IgGs). While these IgGs are not specific to the pathogens present at a farm level and cannot be absorbed by the gut post-weaning due to the established gut structure and junctions, they play a crucial role in regulating the intestinal immune system [17], thereby enhancing the health of piglets after weaning [18].

Another strategic ingredient commonly used in swine production is the yeast *Saccharomyces cerevisiae*, which represents an important source of products with probiotic and prebiotic activity, either when used as a living agent or with compounds derived from its cell wall, respectively [19], in this case highlighting β-glucans, an immunomodulatory component that potentiates immune responses [20].

In this regard, some yeast strains (*Saccharomyces cerevisiae*, for example) and their components may improve the zootechnical performance and immune function of weaner piglets [21,22]. Autolyzed yeast (AY), enzymatically derived from *Saccharomyces cerevisiae*, is characterized by its high concentration of digestible crude protein (41.3% on a dry matter basis) and essential amino acids, including 2.71% lysine and 4.16% glutamate on a dry matter basis [23], mannooligosaccharides, which have prebiotic action, and nucleotides, which are important for the repair and development of rapidly growing tissues, such as the intestinal mucosa [19].

The increasing use of feed additives in animal production, including AY, gained intensity with the prohibition/restriction of the use of antibiotics as growth promoters in 2006, representing a resource to meet the challenges that newly weaned piglets go through [24,25].

Although the use of BP in the diet of weaners represents a resource for providing high amounts of high-quality protein, its high cost often limits the use of higher levels or even inclusion in the diet [26,27]. In this study, rations incorporating plasma were formulated in accordance with commercially recommended inclusion rates, as documented by Remus et al. [27]. The substitution of plasma with autolyzed yeast (AY) was executed at equivalent percentage ratios. Furthermore, the incorporation of an immunomodulator adhered to commercial guidelines. This strategy, combined with the use of AY, enhances the potential for practical application of the findings. This approach is in line with a meta-analytical study by Remus et al. [27] which examined the effects of adding blood plasma to post-weaning piglet diets in terms of the piglets’ productive performance. The innovative approach of this study lies in its application of advanced methodologies to explore the effects of dietary interventions on the intestinal health of early weaned piglets. This includes comprehensive analyses such as microbiome profiling, cytokine response evaluation, and detailed assessments of intestinal absorption and health. Therefore, the objective of this study was to evaluate the effect of the inclusion of AY, replacing BP, as a dietary ingredient on the performance and intestinal health of and immune and metabolic modulation in early-weaned piglets.

## 2. Materials and Methods

### 2.1. Animal Ethics Statement

All procedures adopted in this study were reviewed and approved by the Ethics Committee for Research and Animal Experimentation of Akei Animal Research, number 006/2021.

### 2.2. Sources of Autolyzed Yeast, β-Glucans and Blood Plasm

The autolyzed yeast (AY) used in this study, produced by Hicell^®^ (Biorigin, Lençóis Paulista, SP, Brazil), contained 40.00% crude protein. The profile of digestible amino acids included 1.02% arginine, 1.57% phenylalanine, 0.87% histidine, 2.36% isoleucine, 2.24% leucine, 2.84% lysine, 0.700% methionine, and 0.95% combined methionine and cysteine, 1.52% threonine, and 2.06% valine. The mineral content was 0.11% calcium and 1.50% total phosphorus. Other nutritional components included 8.00% humidity, 3.00% crude fiber, 3.5% ether extract, and 10.0% mineral matter. The energy value was 2812 kcal/kg metabolizable energy for pigs. The immunomodulator β-glucan (Macrocard^®^; Biorigin, Lençóis Paulista, SP, Brazil), derived from yeast cell walls, comprised a minimum of 60.0% 1,3/1,6 β-glucans, with a proportion of 73% 1,3-β-glucans and 27% 1,6-β-glucans.

The plasma used in the study has a detailed nutritional composition as follows: crude protein content stood at 78%. The profile of digestible amino acids included 4.73% arginine, 2% cystine, 2.45% histidine, 2.17% isoleucine, 6.31% leucine, 6% lysine, 0.65% methionine, with a combined methionine and cystine content of 2.68%, 3.77% threonine, 1.02% tryptophan, and 4.1% valine. The mineral content comprised 0.15% total calcium, with a calcium-to-phosphorus ratio of 1.15%, 1.3% total and available phosphorus, 1.1% chlorine, 2.2% sodium, iron at 90 ppm, 0.34% magnesium, and 0.3% potassium. Additional nutritional components included 0.5% crude fiber, 0.3% total fat, 1% non-structural carbohydrate, 9% moisture, and 91% dry matter. The energy content was quantified at 4100 kcal/kg for digestible energy and 3890 kcal/kg for metabolizable energy.

### 2.3. Experimental Design

The experimental design was in randomized blocks (which were formed based on the weaning weight and sex of the piglets), with four treatments and 12 replicates per treatment. The animals received water and feed ad libitum during the experimental period (42 days), and the nutritional program was divided into four phases (pre-initial I and II and initial I and II) based on feed intake and age criteria. The following treatments were applied within the first three nutritional phases: T1 = control diet plus blood plasma (BP); T2 = control diet plus autolyzed yeast (AY); T3 = control diet plus autolyzed yeast and immunomodulator (AYI); and T4 = control diet containing blood plasma (50%) plus autolyzed yeast (50%; BPAY).

### 2.4. Animals, Experimental Unit and Diets

A total of 240 piglets, comprising castrated males (n = 120) and females (n = 120), of the Camborough × AG 337^®^ genetic (Agroceres, Rio Claro, SP, Brazil), were weaned at an age of 21 days. Upon weaning, they had a mean initial weight of 5.247 ± 0.827 kg. The animals were housed in 48 pens with an area of 2.55 m^2^ with a fully slatted floor, pendular nipple drinker (with adjustable height) and linear feeders with six openings. Thermal control of the environment was maintained using a 250-W infrared lamp installed in the center of the pen and side curtains. Temperature and relative humidity were recorded using an Instrutemp ITLOG 80^®^ datalogger (Instrutemp, São Paulo, SP, Brazil). The maximum and minimum air temperature and relative humidity during the experimental period were 26.53 ± 2.59 °C, 25.03 ± 2.50 °C and 70.49 ± 9.54%, respectively.

The feeding program included four phases: pre-initial I (21–28 days of age); pre-initial II (29–35 days of age); initial I (36–42 days of age); and initial II (42–63 days of age) (Table 1). All provided rations were in dry, meal form, primarily consisting of corn and soybean meal, formulated in accordance with the recommendations described by Rostagno et al. [28]. Feed and water were provided ad libitum throughout the experimental period.

Daily weight gain (DWG), daily feed intake (DFI), and the feed conversion ratio (FCR) were evaluated weekly by weighing the piglets individually and calculating their feed intake, with leftovers and waste deducted; all were expressed for each experimental phase and considering the entire evaluation period.

Fresh excreta were ranked using the following scale: 0 = solid; 1 = semi-solid; 2 = semi-liquid; and 3 = liquid [29]. The diarrhea score was calculated by dividing the number of days the animals presented with diarrhea by the total number of days evaluated.

Depression scores were calculated daily and individually using the methodology described by Rossi et al. [30], where 0 indicated animals exhibiting liveliness, alertness and responsiveness; 1 indicated animals that were standing and isolated but quickly responded to stimuli; 2 indicated animals that were standing and isolated, with their head down, and with possible muscle weakness and delayed response to stimuli; and 3 indicated animals that were depressed, lying down, and were reluctant to get up. The depression index was calculated by dividing the number of animals classified under the respective depression scores by the number of total animals evaluated.

The flank score was determined using the method described by Spiehs, Shurson, & Johnston [31]: a score of 1 indicated an animal with a normal abdomen and full and round flanks, a score of 2 indicated an animal with a slightly full intestine and flat flanks, and a score of 3 indicated a severely lean animal and empty flanks. The flank index was calculated by dividing the number of animals identified in the respective flank scores by the number of total animals evaluated. The administered medications in this study included both antibiotics and anti-inflammatory drugs, targeting a spectrum of piglet health issues such as enteric and respiratory diseases, as well as inflammatory conditions. This therapeutic approach was integral to managing the diverse health challenges encountered during the experimental period.

### 2.5. Intestinal Permeability Evaluation and Blood Sampling

At 32 days of age (Day 11), twelve piglets per treatment group (one animal per pen, totaling 48) were orally administered Fluorescein Isothiocyanate Dextran (FITC-dextran) (3–5 kD; Sigma–Aldrich^®^, St. Louis, MO, USA), at a dose of 25 mg per pig (1 mL of solution), following a 6 h fasting period. This administration was conducted to assess intestinal permeability using the method outlined by Vicuña et al. [32]. Subsequently, 2 h and 30 min post-administration, blood samples were collected via jugular puncture.

At 35 (D14) and 49 (D28) days of age, 48 piglets were randomly selected (12 per treatment, 6 of each sex) for blood collection from the jugular vein. The D14 samples were analyzed using a commercial kit (Invitrogen™, Waltham, MA, USA, EPX090-60829-901), and the Luminex ×MAP technique was employed to determine the concentration of Interleukin (IL)-1β, IL-4, IL-6, IL-8, IL-10, IL-12p40, interferon alpha (IFN-α) and gamma (IFN-γ) and tumor necrosis factor alpha (TNFα). The samples from D14 and D28 were analyzed to determine the amount of glucose, triglycerides, cholesterol, insulin, leptin, beta hydroxybutyrate (BHB) and non-esterified fatty acids (NEFAs) they contained.

### 2.6. Gut Microbiota

On day 14 of the experiment, fecal samples were collected via rectal swab from 12 animals per treatment group, amounting to a total of 48 animals (1 animal per pen); the samples were immediately transferred to individual Eppendorf tubes and frozen at −80 °C; subsequently, the samples were submitted for large-scale bacterial identification and bacterial counts by DNA sequencing.

Bacterial DNA was extracted using a ZR Fecal DNA MiniPrep^®^ kit from Zymo Research (Zymo Research, Murphy Ave., Irvine, CA, USA) following the protocol recommended by the manufacturer. The extracted DNA was quantified by spectrophotometry at 260 nm. To evaluate the integrity of the extracted DNA, all samples were analyzed by electrophoresis on a 1% agarose gel.

A ~460-base segment of the V3V4 hypervariable region of the 16S ribosomal gene was amplified using universal primers and the following PCR conditions: 95 °C for 3 min; 25 cycles of 95 °C for 30 s, 55 °C for 30 s and 72 °C for 30 s; and 72 °C for 5 min [33]. A metagenomic library was constructed from the amplicons using the “Nextera DNA Library Preparation Kit” from Illumina^®^ (San Diego, CA, USA). The amplicons were pooled and subsequently sequenced in an Illumina^®^ “MiSeq” sequencer [33].

Pair-end reads obtained from the sequencer were analyzed using the QIIME2 (Quantitative Insights Into Microbial Ecology) platform [34,35]. This was followed by a workflow that included the removal of low-quality sequences, filtration, the removal of chimeras and taxonomic classification. The sequences were classified into bacterial genera through the recognition of amplicon sequence variants (ASVs), in this case, the homology between the sequences when compared against a database [36]. The 2021 update (GTDB 202) of the ribosomal sequences database Genome Taxonomy Database was used to compare the sequences [36]. To classify bacterial communities by ASV identification, 14,994 readings were used per sample to normalize the data and to ensure that each sample had a similar number of readings.

### 2.7. Statistical Analyses

The normality of the distribution of the data was analyzed using the Kolmogorov–Smirnov and Lilliefors test and the Shapiro–Wilk W test (*p* > 0.05). The Box and Whisker package was used to remove outliers. Data following a normal parametric distribution underwent analysis of variance using the General Linear Model (GLM), with the model considering block, time, and treatment effects. The means from this analysis were further evaluated using Tukey’s test. For performance data, except in the first week, live weight was used as a covariate in the analysis. Additionally, for categorical data, both the GLM and the chi-square test were utilized. Both analyses were performed using Statistic for Windows^®^ software, version 10.0 (StatSoft; Tulsa, OK, USA). For the tests, a *p*-value equal to or less than 0.05 was considered significant, and a *p*-value between 0.05 and 0.10 was considered a trend.

Statistical comparisons between the alpha diversities for each analyzed group were conducted using the nonparametric Kruskal–Wallis test and Dunn’s post-test, with results less than 0.05 considered statistically significant. The analyses for beta diversity were performed using perMANOVA in the QIIME2 pipeline, using 10,000 permutations. All figures and statistical analyses were performed in “R”. Alpha diversity was calculated using the “phyloseq” [37], “vegan [38] ” and “Microbiome” [39] packages. The differences in the relative abundances of the taxa between the analyzed groups were estimated using the Kruskal–Wallis test and the Dunn post hoc test. 

## 3. Results

### 3.1. Performance

Table 2 presents the zootechnical performance results for the experimental phases and considers the all-nursery period. In pre-initial phase I (21–28 days), there was no difference between the analyzed parameters; in pre-initial phase II (29–35 days), for piglets in the AY group, there was an improvement in the FCR of 8.49% (0.126), 1.52% (0.021) and 4.50% (0.084) compared with the FCR in the BP, AYI and BPAY groups, respectively (*p* ≤ 0.05). For the other parameters analyzed in the pre-initial II and initial I phases (36–42 days), no differences were found among treatments. In the first three weeks after the weaning (21–42 days) periods in which the experimental treatments were applied, the animals in the AY and AYI groups showed decreased FCR (*p* ≤ 0.05) compared to the animals in the BP group, i.e., 4.64% (0.071) and 4.58% (0.070), respectively. In the evaluation of the all-nursery period, including initial phase II, no differences were observed among treatments for the performance parameters evaluated.

### 3.2. Intestinal Health and Blood Parameters

There were no differences among treatments regarding the concentrations of IFN-α, IFN-γ, IL-10, IL-1β, IL-4, IL-6, IL-8, TNF-α, and IL-12p40, and there were no changes in the intestinal permeability of piglets fed or not fed AY associated with the immunomodulator (AYI) or with plasma (BPAY) (Table 3). Table 4 shows the diarrhea and thinness scores and indices as well as the number of medicated animals and medications used during the total study period. For piglets fed AY and BPAY, there was an 82.35% (−14n) and 88.23% (−15n) reduction in the number of medications, respectively, compared with the number of medications in the BP group (17n; *p* ≤ 0.05). For the diarrhea score in all periods of the test (scores 2 and 3), there was a trend (*p* = 0.0723) of lower occurrence in the AYI group. There was no difference in the thinness score and use of medication among treatments, and no animals exhibited depression as a side effect of any of the treatments.

At 35 days of age, piglets fed AY showed differences in the concentrations of triglycerides and beta hydroxybutyrate (BHB) (*p* ≤ 0.05; Table 5). In piglets in the AY, AYI, and BPAY groups, there was a 79.32%, 69.83% and 69.00% increase in triglycerides, respectively, compared to the triglyceride level in the BP group (*p* ≤ 0.05). The concentration of BHB was 31.95%, 22.64% and 5.89% lower in the AY group than in the BP, AYI and PBAY groups, respectively (*p* ≤ 0.05). For glucose, cholesterol, insulin, leptin, and non-esterified fatty acids (NEFAs), there were no differences between the groups analyzed. As in the second collection, at 49 days of age, there were no differences among treatments for all parameters analyzed.

### 3.3. Gut Microbiota

The alpha diversity (Figure 1) analysis was performed using the Chao1 (Figure 1A), observed OTUs (Figure 1B), Fisher (Figure 1C), Simpson (Figure 1D), Shannon (Figure 1E) and Evenness Pielou (Figure 1F) indices. There were no differences among groups for any of the tested metrics. Beta diversity (Figure 2) was estimated using the Bray–Curtis (Figure 2A; *p* = 0.257074), Jaccard (Figure 2B; *p* = 0.80472), UniFrac (Figure 2C; *p* = 0.526947), and weighted UniFrac (Figure 2D; *p* = 0.262274) parameters. There were no differences in the dissimilarity of the taxa present, estimated by all metrics, among the four groups.

Regarding the taxonomic composition of the bacterial community observed 14 days post weaning, the most abundant phyla identified in the samples included *Firmicutes*, *Actinobacteriota* and *Bacteroidetes*. The taxa that were different were members of the *Acutalibacteraceae* family, differing between piglets in the BP and AYI groups (*p* ≤ 0.05), with the lowest and highest abundances, respectively, and members of the *Coriobacteriaceae* family (Figure 3B), with lower abundance in the AY group than in BP and AYI groups (*p* ≤ 0.05). Regarding genera, *Collinsella* (Figure 4A) was less abundant in the AY group than in the BP and AYI groups (*p* ≤ 0.05), and the abundance of *Lactobacillus* (Figure 4B) was different between the AY and AYI groups (Figure 4B).

The *Firmicutes*/*Bacteroidota* (F/B) ratio was calculated for each sample analyzed, and no differences were found between the groups tested.

At the species level, *Bulleidia* sp900539965 (Figure 5A) was less abundant in the AY group than in the AYI and BPAY groups (*p* ≤ 0.05). *Clostridium saudiense* (Figure 5B) was more abundant in piglets in the BP group than in piglets in the AY and AYI groups (Figure 5B). Finally, *Collinsella* sp002391315 (Figure 5C) abundance was different between the AY group and the BP and AYI groups (Figure 5C).

## 4. Discussion

### 4.1. Performance

Recent studies have shown positive effects of the use of yeast (*Saccharomyces cerevisiae*) and its derivatives on the zootechnical performance of weaner piglets [8,19,40,41,42]. The total or partial replacement of BP with AY (associated or not with the immunomodulator, AYI) in pre-initial phase I did not result in differences in DWG, DFI, and FCR (Table 2). There was only an increasing trend (*p* = 0.0970) for DFI for piglets fed BP, reiterating the virtue of this ingredient regarding the palatability that it confers to the diets [17,43], which, however, was restricted to the first week after weaning.

Although a higher feed intake in the first week after weaning has been shown to have impacts on intestinal health and increase the body weight of piglets [44], these findings were not observed in the present study. Additionally, such consumption results are inconsistent; Shurson [45] highlights that AY generates positive effects, a finding opposed by Boontiam et al. [8], who observed a worsening in feed intake when the inclusion of AY increased from 5% to 10% in the diets of piglets between 15 and 28 days of age, with a quadratic response for the parameter, evidencing its negative effect only at levels higher than the 5% inclusion rate, an effect which was not observed in our study.

In pre-initial phase II (Table 2), piglets fed AY (5%) presented better FCR (*p* ≤ 0.05). The addition of between 1.16 and 1.64% yeast extract in the diet of piglets aged between 21 and 35 days old, replacing 50% of the BP in the diet, improved the structure of the intestinal mucosa (duodenal villi height and depth of the crypt) [46], a result that, according to the authors, had a positive influence on FCR, a fact that may support our results. In agreement with this finding, performance (LW and DWG), with no change in FCR, improved in piglets fed diets with three different levels of free nucleotides (hydrolyzed dehydrated *Kluyveromyces fragilis* yeast extract) compared with those for piglets fed a negative control diet (nucleotide free) [47]. This finding reiterates the benefits of AY on feed efficiency.

Waititu et al. [48], working with groups treated and untreated with antibiotics as growth promoters associated or not with diets with three levels of inclusion of nucleotides (0, 1000 or 2000 ppm), observed that yeast extract also improved the DWG of weaned piglets. The performance results obtained in the present study can be supported by the findings described above, which depict AY as a performance-enhancing ingredient (Table 2).

In the evaluation of the period in which the piglets were fed the additives/ingredients evaluated (21–42 days of age), the FCR of animals in the AY and AYI groups was better than that in the BP group (*p* ≤ 0.05) and similar to that in the BPAY group. This condition of better feed efficiency is supported by the results obtained by Cruz et al. [49], who used the yeast *Candida utilis* as a protein source, replacing the crude protein of the experimental diets of piglets weaned at 30 days of age at 10, 20 or 40%, and observed better intestinal performance and function, highlighting the digestibility and mucosal morphometry of such diets. Garcia et al. [42], when evaluating three levels of *Saccharomyces cerevisiae* (1 × 10^7^ CFU/g of feed) as a protein source in the diet of piglets weaned at 21 days of age, found that the inclusions, compared with the control diet, promoted better performance and immune response, highlighting the ability of the yeast strain to control intestinal inflammation associated with early weaning, in addition to improvements in FCR.

In a comparative study with piglets weaned at 19 days of age subjected to three treatments, i.e., a negative control diet without plasma or yeast extract and two test diets, one with 5% (supplied during days 1 to 14 of the study) and 2.5% plasma (supplied during days 14 to 28 of the study) and another with yeast extract (same doses and inclusion periods) [50], advantages were identified for crypt depth and intestinal wall thickness for the test diets, and there were better results for the villus width and lamina propria area for the diet with yeast compared with the control diet, in addition to better DWG, effects that were maintained until the fattening stages, especially for the diet with yeast extract.

Using yeast cell wall exclusively compared with a diet with growth-promoting antibiotics, Qin et al. [51] observed that piglets weaned at 23 days fed yeast glycoprotein (800 mg/kg of feed) had a higher live weight, increased DWG and better FCR, in addition to better intestinal development, strengthening the intestinal health of piglets. This result indicates that the AYI group had the best FCR and the best results for the entire experimental period, in which the prebiotic role of this component favored the reduction in the expression of proinflammatory cytokines (Table 3) and the lowest occurrence of diarrhea (Table 4).

Notably, BP is a source of quality protein and vitamin and mineral composition, especially vitamin B12, calcium, phosphorus, sodium, chloride, potassium, and magnesium [17], in addition to having high levels of immunoglobulins (IgGs), which play roles in the regulation of the immune system, providing health benefits to recently weaned piglets [14,17,52]. Yeast and its products, AY and cell wall, associated with or partially replacing BP, showed advantages in some phases, or at least indices that were similar to those obtained using BP, demonstrating its value as an ingredient that has high digestibility [53] and palatability [45] and that effectively participates in performance and passive and active immunity, reducing pathogenic loads and the occurrence of postweaning diarrhea, modulating beneficial microbial growth, and stimulating the development of the gastrointestinal epithelium [48,50,53].

Additionally, it can be inferred that AY and its combinations offer an economically viable option for production systems aiming to obtain lower-cost feeds with significant zootechnical benefits. This is particularly relevant given the recognized drawback of BP’s high price [26], which sometimes constrains its usage and inclusion levels in diets. Notably, AY costs approximately 20–25% of the price of BP, further emphasizing its cost-effectiveness.

### 4.2. Intestinal Health and Blood Parameters

The results of the analysis of plasma cytokines indicated that the immunological status of piglets did not differ among treatments. However, two trends were detected for IFN-α (*p* = 0.0693) and IL-10 (*p* = 0.0854), results that are in agreement with those found in mesenteric lymph nodes of newly weaned piglets fed diets containing yeast components, such as β-glucans, in which there were higher levels of the anti-inflammatory cytokine IL-10 [41].

These results suggest that yeast products induce or maintain an anti-inflammatory state in the immune system of piglets. In this context, yeast products may alter secondary immune responses [54], as shown by the increase in IL-10 production after 2 and 4 h of infection with *L. braziliensis* in macrophages. Boontiam et al. [8] oobserved that AY supplementation in piglets reduced the proinflammatory cytokines IL-1, IL-6, and TNF-α.

Pigs weaned early and fed a diet without additives are susceptible to proinflammatory cytokine secretion [8,41]. The lower concentration of cytokines in this study compared to the concentration of cytokines reported in studies by Boontiam et al. [8] and de Vries et al. [41] can be explained by the quality of the ingredients used in all treatments (BP, AY, AYI, and BPAY), considering that all have recognized actions in immune modulation [14,17,51,52].

Proinflammatory cytokines, at high concentrations, may contribute to the onset of diarrhea in piglets [55]. The number of cases of diarrhea with scores of 2 and 3 was not significantly different among the groups evaluated (Table 4); however, there was a trend (*p* = 0.0816) for diarrhea with scores of 2 and 3, with the animals in the AYI group presenting a lower number of cases. Prebiotics are known to help maintain intestinal health, as is the case with oligosaccharides, which minimize intestinal inflammation by modulating the microbiota or influencing the expression of cytokines [56]. This observation is consistent with our findings, considering that the AYI group was the only one that received immunomodulator (0.025%). Additionally, the similarity in the number of cases of diarrhea between the AY and AYI groups and the BP group indicates that partial or total replacement of BP with yeast extract in the diets positively affected the intestinal disorders evaluated.

Early weaning in piglets is challenging due to the high probability of diarrhea in the first 14 days after separation. However, Boontiam et al. [8] found that the use of AY in the diets of weaners, between 0 and 10%, linearly decreased the occurrence of this condition, an effect that may be associated with the better digestibility of nutrients conferred by AY, leading to the rapid recovery of the intestinal mucosa, which is reflected in the performance and in the minimization of digestive problems, as observed in our results. AY also contains nucleotides, which act in the repair and development of rapidly growing tissues, such as the intestinal mucosa, especially in challenging conditions such as weaning [23,57].

Regarding the role of glycoprotein, an immunomodulator present in the AYI treatment, its actions are greatly amplified, which supports the positive results regarding the occurrence of observed diarrhea (Table 4). Qin et al. [51], using a diet supplemented with yeast glycoprotein (800 mg/kg of feed) compared to a diet with antibiotics at the promoter level, observed that piglets weaned at 23 days of age and fed the glycoprotein diet showed improvements in intestinal permeability and intestinal development. At the histological level, there were increases in villus height and in the ratio of villus height to crypt depth and decreases in crypt depth and villus width in the ileum. Additionally, intestinal integrity was facilitated by the glycoprotein diet, as there was an upregulation of occludin mRNA expression in the duodenal and jejunal mucosa of weaned piglets.

Regarding cytokines [51], glycoprotein supplementation negatively regulated IL-12 m-RNA expression and positively regulated Hsp-70 m-RNA expression in the duodenal mucosa, negatively regulated Hsp-70 mRNA expression and positively regulated IFN-γ and Hsp-90 m-RNA in the jejunal mucosa, and positively regulated the expression of Hsp-70 m-RNA in the ileal mucosa. Additionally, in addition to improvements in intestinal health, there was an increase in the relative abundance of the genus *Lactobacillus*.

AY containing IFN-α, an anti-inflammatory cytokine, demonstrated superior results compared to AYI and comparable outcomes to both BP and the combination of BP with AY. This is indicative of the immunomodulatory potential of these ingredients, as supported by Pérez-Bosque et al. [18] and Hu et al. [58]. The lower IFF-α levels observed in the AYI treatment correlate with reduced diarrhea incidence, which is sometimes attributed to viral factors. Furthermore, the influence of AY, with or without an immunomodulator, on IL-10 levels, particularly in combination with BP, indicates a dosage effect. The lower AY dosage appeared more effective in modulating this anti-inflammatory cytokine. The enhanced outcomes of BPAY over BP align with the findings of Hu et al. [58], affirming AY’s superior immunomodulatory efficacy compared to BP.

In this study, injectable therapeutic interventions included the use of both antibiotics and non-steroidal anti-inflammatory drugs (NSAIDs). Upon comparative analysis, it was observed that piglets in the groups receiving AYI, and those receiving a combination of blood plasma and autolyzed yeast (BPAY), showed more favorable health outcomes compared to the group receiving only BP (*p* ≤ 0.05; Table 4). The increased frequency of medication in the BP-treated piglets may impact the integrity of the intestinal barrier, subsequently affecting digestive functions. Saladigas-Garcia et al. [59] reported that weaned piglets experiencing health challenges exhibited downregulation of genes associated with intestinal health, leading to decreased triglyceride levels and elevated serum BHB, corroborating our findings. The presence of an immunomodulator, such as β-glucans, which constitute 30–60% of the total cell wall polysaccharides [8,57,60], improves the functional status of macrophages and neutrophils [54,61], modulates immunosuppression [8,20,56], okand increases resistance to infections by Gram-negative bacteria [21], antioxidant capacity [62], and antitumor activity [60]. In addition to the functions already listed, the results of this study reveal the mechanisms of action of the ingredients/additives evaluated.

At 35 days of age, piglets fed AY had higher amounts of triglycerides than piglets in the BP group (Table 5; *p* ≤ 0.05). There is a limited number of studies that have evaluated the blood parameters and the metabolic profile of piglets in the nursery phase fed diets containing AY and BP. Czech et al. [40] observed that the inclusion of yeasts (*Yarrowia lipolytica* or *Saccharomyces cerevisiae*) in the diet of weaners promoted an increase in glucose and high-density lipoprotein (HDL) and a decrease in uric acid and total and low-density lipoprotein (LDL) cholesterol. Zhang et al. [53] reported that yeast extract decreased the blood cholesterol content of swine, results which were not observed in our study.

Triglycerides represent an energy store of yellow adipose tissue, which, through hydrolysis (lipolysis), can be mobilized or stored based on the demands of peripheral tissues, such as the liver and cardiac and skeletal muscles. However, they can also be supplied by food, thus increasing their concentration in the blood. In this regard, it is possible to associate higher triglyceride values for the AY, AYI and BPAY groups with better perceived BP values for these treatments, indicating the better absorption of nutrients, including lipid components. However, unlike the results obtained in our study, Hu et al. [22], in a study of piglets weaned at 21 days of age and fed diets with 4% plasma, 2% plasma plus 2% protein and 4% protein derived from yeast, did not identify differences in triglyceride levels at 7 days post-weaning, attributing these results to the similar feed intake of the experimental groups. This result [22], however, can be considered positive, as the diets with yeast components were equivalent to a diet with an ingredient (BP) of recognized nutritional quality.

Beta-hydroxybutyrate (BHB), found in higher concentrations in the group of piglets fed BP (Table 5; *p* ≤ 0.05) than in the piglets fed AY and AYI, is a serum ketone produced in the liver and directed to peripheral tissues for use as a source of energy, with the highest levels coming from the degradation of hepatic fatty acids, which occurs in the liver as a result of reduced caloric intake due to fasting or anorexia [63].

Higher BHB was not observed in any of the experimental groups, as confirmed by the zootechnical indices obtained (Table 2). Regarding the nutritional characteristics of the diets, high-fat and low-carbohydrate diets tend to increase ketone bodies because the increase in BHB results from the degradation of fatty acids in the liver and the reduction in blood glucose levels [64]. This condition, however, also was not observed in the results obtained; the rations were all isonutrients and isocaloric and the intake was similar among groups (Table 2). Thus, further studies are needed to elucidate the mechanism by which AY triggers energy metabolism in piglets.

### 4.3. Gut Microbiota

Long-chain (n-3) fatty acids [65] and symbiotics [66] are additives that have profiles that can replace APCs [67] and are able to preserve the intestinal health of piglets via the maintenance of alpha and beta diversity without causing intestinal dysbiosis, an effect observed in all groups of this study for both diversities (Figure 1 and Figure 2).

The predominant bacterial phyla in the swine intestinal microbiota include Firmicutes, *Proteobacteria,* and *Bacteroidetes* [68,69]. This profile is similar to the profile observed in the present study, however, there was a lower relative abundance of *Proteobacteria* than *Actinobacteriota*, which was among the three most abundant phyla. Despite this difference, the finding tends to be positive because the phylum Proteobacteria is abundant in cases of intestinal dysbiosis [70], and it is described by Shin et al. [71] as an agent responsible for determining these conditions. The *Firmicutes*/*Bacteroidota* ratio is related to the maintenance of homeostasis [72] and to weight gain [73], supporting the zootechnical results obtained (Table 2); that is, all groups without plasma were similar to the group that received plasma in their diet.

For the AYI group, the family *Acutalibacteraceae* was present in greater relative abundance than in the other groups, being different only in relation to the BP group (Figure 3A). However, there is limited information regarding the relevance of this family to the swine intestinal microbiota, which was recently described [74,75,76].

The AYI group also showed the highest relative abundance of the *Coriobacteriaceae* family among all the treatments evaluated, followed by the BP group. The relative abundance of the *Coriobacteriaceae* family in both groups was different from that in the AY group (Figure 3B). The same results were found for the genus *Collinsella* (Figure 4A) and the species *Collinsella* sp002391315 (Figure 5C), indicating that the modulation that occurred at the species level significantly modified the relative abundance of the higher direct taxa up to the family level.

*Coriobacteriaceae* is a component of different mammalian microbiota (oral, gastrointestinal, and genital), and its representatives, 30 species belonging to 14 genera, may act in the metabolism of bile salts and steroids and in the activation of food polyphenols, such as in the bioactivation of daidzein [77]. This result was observed with the additive used exclusively in AYI treatment, an immunomodulator, whose premise for use is the modulation of the intestinal microbiota [19,56]. The genus *Collinsella* has been described as being positively correlated with the concentration of isovaleric acid in the intestine [78]. In a study that evaluated the modeling of human enteric dysbiosis and rotavirus immunity in gnotobiotic pigs, Twitchell et al. [79] identified a strong correlation between this genus and the responses of IFN-γ-producing CD8+ T cells specific to intestinal and circulating rotavirus, which are correlated with protection against rotavirus-related diarrhea. This finding may support the better results obtained by the AYI treatment regarding the incidence of diarrhea, which may also be due to the immunomodulator used in this treatment.

Predominant in the postweaning intestinal microbiota [80], the genus *Lactobacillus* spp., which is in greater abundance in the AYI group than in the AY group (Figure 4B), is directly related to improved feed efficiency and increased weight gain in swine [81,82]. *Lactobacillus* are known for their production of lactic acid, which has bactericidal action [83], and direct effects on the intestine, activating cellular components of the innate immune system such as intestinal epithelial cells, natural killer (NK) cells, and monocytes [84]. According to Ricke et al. [85], bacteria of this genus ferment prebiotics, suggesting that the presence of prebiotics may affect the use of amino acids by the host. Khan et al. [86] also reported a positive correlation of the presence of prebiotics with an increase in the relative abundance of bacteria of this genus. Thus, the addition of prebiotics to the diets of animals in the AYI group may have favored bacteria of the genus *Lactobacillus*.

*Bulleidia* sp900539965 was lower in the AY group than in the AYI and BPAY groups (Figure 5A). Bacteria of this genus are Gram-positive bacilli of the phylum *Firmicutes* and are related to *Treponema*, *Ruminococcus*, *Collinsella*, *Fibrobacter*, *Phascolarctobacterium*, *Rummeliibacillus*, *Butyricicoccus*, *Oribacterium*, *Blautia*, *Sphaerochaeta*, and *Peptococcus*, with higher reproductive performance in animals with low performance. They did not appear to have an effective relationship with the parameters evaluated in this study.

Finally, *Clostridium saudiense* was significantly abundant in the animals in the BP group in relation to the animals in the AY and AYI groups (Figure 5B). This species, which is almost always present in the intestinal microbiota of animals in the production chain, does not have a widely known specific function in the gastrointestinal tract of pigs, and thus requires further investigation [87]. However, the presence of bacteria of this species isolated from the digestive tract of swine was negatively correlated with average daily gain, carcass weight, and butyrate concentration [88], which suggests that plasma can be replaced with autolyzed yeast at the same level of inclusion in diets. It is important to note that these observations pertain to the period 14 days post-weaning, focusing on the immediate recovery from weaning stress. However, it is acknowledged that additional changes in the microbial community might have emerged had they been observed at the conclusion of the trial.

## 5. Conclusions

This study conclusively demonstrates the effectiveness of autolyzed yeast, both as a standalone ingredient and in combination with immunomodulators (β-glucans 1,3 and 1,6), as a practical substitute for blood plasma in the diets of nursery-phase piglets. Our findings prominently feature improved feed-conversion ratios, a significant reduction in the need for medications, and beneficial modifications in immune response, metabolic parameters, and gut microbiota. These results underscore the utility of ingredients in enhancing piglet nutrition and suggest wider applications in cost-efficient, health-optimized swine-rearing practices.

## Figures and Tables

**Figure 1 animals-14-00631-f001:**
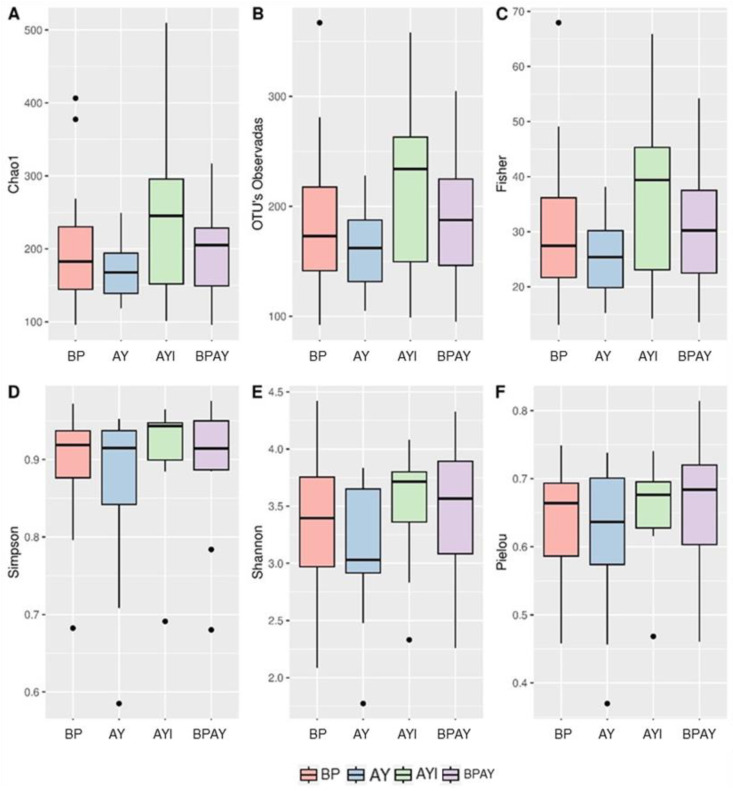
Alpha diversity estimated by the Chao1 (**A**), observed OTUs (**B**), Fisher index (**C**), Simpson index (**D**), Shannon index (**E**) and Evenness Pielou (**F**) indices. Comparisons between treatments were performed using the Kruskal–Wallis test and Dunn’s post hoc test (*p* ≤ 0.05). T1 = control diet plus blood plasma (BP); T2 = control diet plus autolyzed yeast (AY); T3 = control diet plus autolyzed yeast and immunomodulator (AYI); T4 = control diet containing blood plasma (50%) plus autolyzed yeast (50%; BPAY).

**Figure 2 animals-14-00631-f002:**
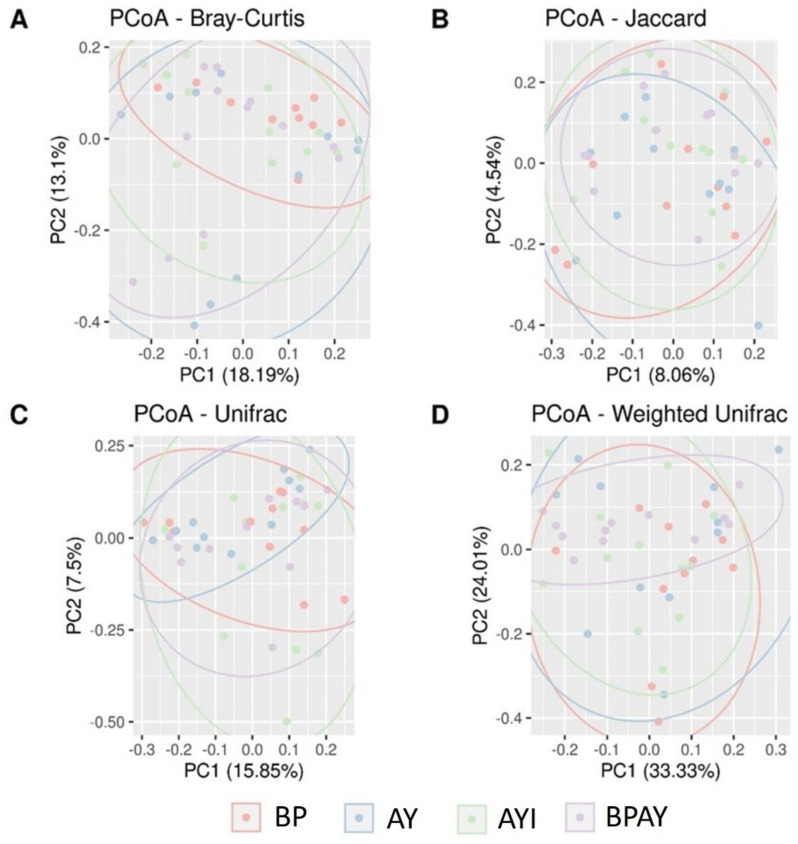
Beta diversity estimated based on the Bray–Curtis (**A**), Jaccard (**B**), UniFrac (**C**) and weighted UniFrac (**D**) parameters. Colored ellipses were automatically added to R via the ggforce library. BP = control diet plus blood plasma; AY = control diet plus autolyzed yeast; AYI = control diet plus autolyzed yeast and immunomodulator; BPAY = control diet containing blood plasma (50%) plus autolyzed yeast (50%).

**Figure 3 animals-14-00631-f003:**
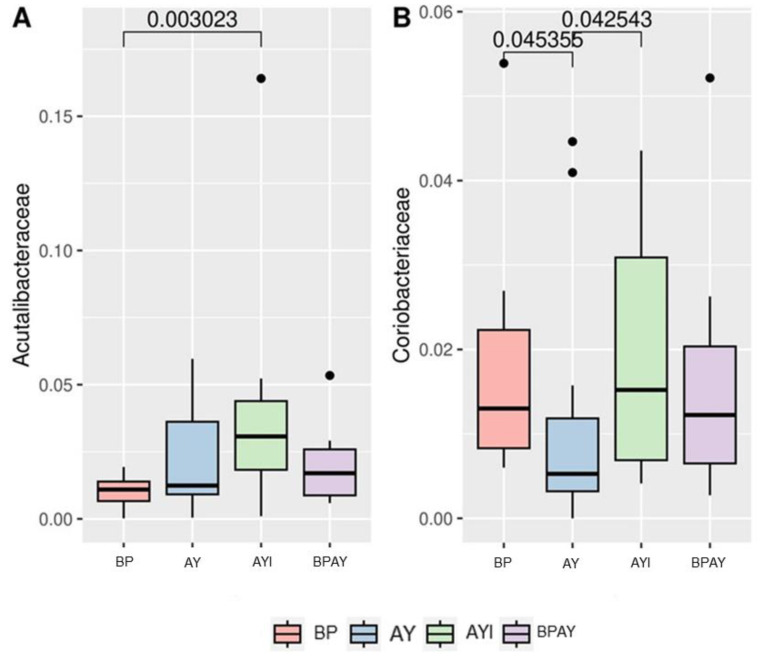
Differential abundance of the *Acutalibacteraceae* (**A**) and *Coriobacteriaceae* (**B**) families. Comparisons between treatments were performed using the Kruskal–Wallis test and Dunn’s post hoc test (*p* ≤ 0.05). T1 = control diet plus blood plasma (BP); T2 = control diet plus autolyzed yeast (AY); T3 = control diet plus autolyzed yeast and immunomodulator (AYI); T4 = control diet containing blood plasma (50%) plus autolyzed yeast (50%; BPAY).

**Figure 4 animals-14-00631-f004:**
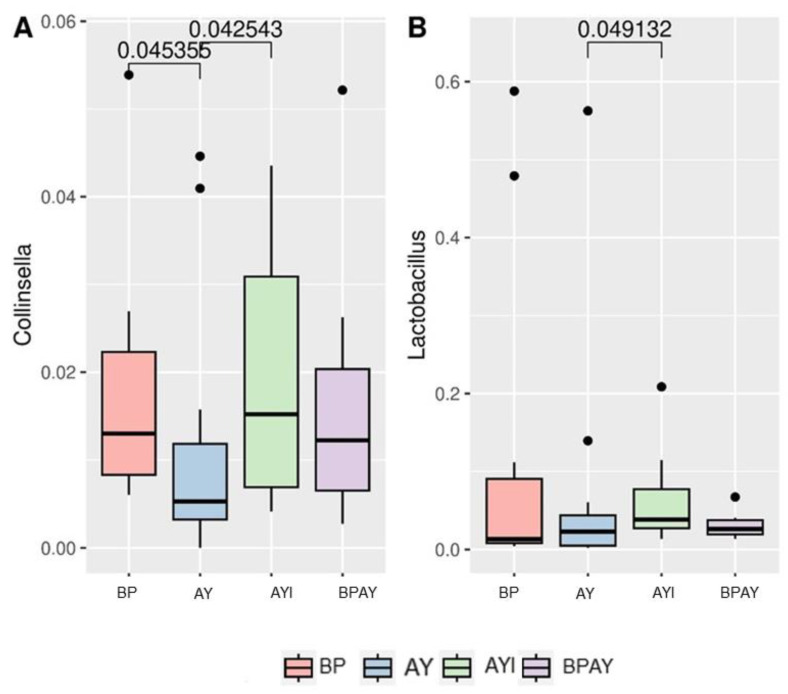
Differential abundance of the genera *Collinsella* (**A**) and *Lactobacillus* (**B**). Comparisons between treatments were performed using the Kruskal–Wallis test and Dunn’s post hoc test (*p* ≤ 0.05). T1 = control diet plus blood plasma (BP); T2 = control diet plus autolyzed yeast (AY); T3 = control diet plus autolyzed yeast and immunomodulator (AYI); T4 = control diet containing blood plasma (50%) plus autolyzed yeast (50%; BPAY).

**Figure 5 animals-14-00631-f005:**
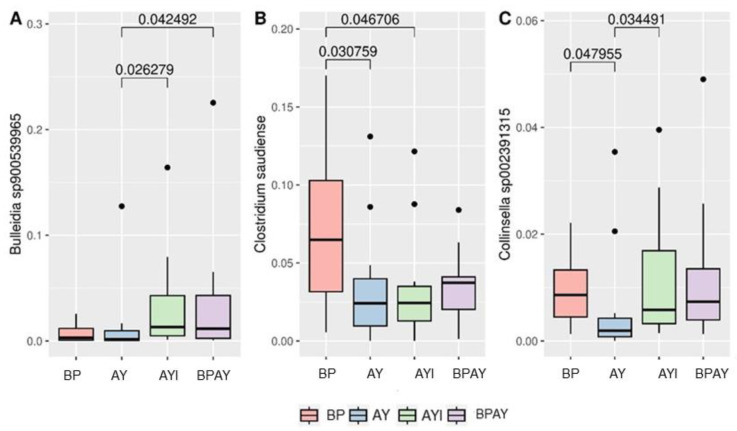
Differential abundance of *Bulleidia* sp900539965 (**A**), *Clostridium saudiense* (**B**) and *Collinsella* sp002391315 species (**C**). Comparison between treatments was performed using the Kruskal–Wallis test and Dunn’s post hoc test (*p* ≤ 0.05). T1 = control diet plus blood plasma (BP); T2 = control diet plus autolyzed yeast (AY); T3 = control diet plus autolyzed yeast and immunomodulator (AYI); T4 = control diet containing blood plasma (50%) plus autolyzed yeast (50%; BPAY).

**Table 1 animals-14-00631-t001:** Composition of and nutritional values for the experimental diets for the different treatments and stages.

Ingredients (%)	Pre-Starter I				Pre-Starter II				Starter I				Starter II
	BP	AY	AYI	BPAY	BP	AY	AYI	BPAY	BP	AY	AYI	BPAY	
Corn	45.37	40.77	40.77	43.04	42.71	39.91	39.91	41.31	50.99	49.56	49.56	50.27	62.27
Soybean meal (45%)	10.50	10.50	10.50	10.50	20.00	20.00	20.00	20.00	27.05	28.03	28.03	27.54	29.32
Pre-cooked corn	10.00	10.00	10.00	10.00	10.00	10.00	10.00	10.00	5.00	5.00	5.00	5.00	-
Whey	14.00	14.00	14.00	14.00	12.00	12.00	12.00	12.00	7.00	7.00	7.00	7.00	-
Star Pro ^3^	10.00	10.00	10.00	10.00	5.00	5.00	5.00	5.00	2.50	2.50	2.50	2.50	-
Plasma	5.00	-	-	2.50	3.00	-	-	1.50	1.00	-	-	0.50	-
Autolyzed yeast	-	5.00	5.00	2.50	-	3.00	3.00	1.50	-	1.00	1.00	0.50	-
Soy oil	1.07	2.00	2.00	1.56	2.27	2.86	2.86	2.57	-	-	-	-	3.94
Soy protein concentrate	0.33	3.54	3.54	1.94	1.48	3.41	3.41	2.45	2.28	2.59	2.59	2.44	-
Dicalcium phosphate	0.86	1.19	1.19	1.03	0.92	1.12	1.12	1.02	1.36	1.42	1.42	1.39	1.75
Limestone	0.22	0.01	0.01	0.11	0.63	0.50	0.50	0.56	0.63	0.58	0.58	0.60	0.85
Immunomodulator	-	-	0.03	-	-	-	0.03	-	-	-	0.03	-	-
L-lysine	0.60	0.70	0.70	0.65	0.40	0.46	0.46	0.43	0.46	0.48	0.48	0.47	0.47
L-threonine	0.31	0.38	0.38	0.34	0.17	0.21	0.21	0.19	0.20	0.22	0.22	0.21	0.20
DL-methionine	0.28	0.34	0.34	0.31	0.20	0.24	0.24	0.22	0.21	0.22	0.22	0.21	0.15
L-valine	0.28	0.35	0.35	0.32	0.08	0.13	0.13	0.11	0.03	0.05	0.05	0.04	0.08
L-tryptophan	0.05	0.10	0.10	0.08	0.02	0.05	0.05	0.03	0.04	0.05	0.05	0.04	0.03
Adsorbent	0.15	0.15	0.15	0.15	0.15	0.15	0.15	0.15	0.15	0.15	0.15	0.15	0.15
Salt	0.32	0.32	0.32	0.32	0.32	0.32	0.32	0.32	0.44	0.50	0.50	0.47	0.53
Banox ^4^	0.01	0.01	0.01	0.01	0.01	0.01	0.01	0.01	0.01	0.01	0.01	0.01	0.01
Zinc oxide	0.31	0.31	0.31	0.31	0.31	0.31	0.31	0.31	0.31	0.31	0.31	0.31	-
Copper sulphate	0.08	0.08	0.08	0.08	0.08	0.08	0.08	0.08	0.08	0.08	0.08	0.08	-
Vitamin Premix ^1^	0.15	0.15	0.15	0.15	0.15	0.15	0.15	0.15	0.15	0.15	0.15	0.15	0.15
Mineral Premix ^2^	0.10	0.10	0.10	0.10	0.10	0.10	0.10	0.10	0.10	0.10	0.10	0.10	0.10
Nutrients													
Metabolic energy (kcal/kg)	3550	3547	3547	3550	3460	3460	3460	3460	3350	3350	3350	3350	3350
Protein, %	18.50	18.50	18.50	18.50	20.00	20.00	20.00	20.00	20.00	20.00	20.00	20.00	19.45
Fat, %	3.15	3.99	3.99	3.59	4.32	4.85	4.85	4.58	4.55	4.84	4.84	4.69	6.46
Crude fiber, %	1.68	1.87	1.87	1.77	2.24	2.35	2.35	2.29	2.68	2.74	2.74	2.71	2.95
Calcium, %	0.53	0.53	0.53	0.53	0.68	0.68	0.68	0.68	0.75	0.75	0.75	0.75	0.87
Total phosphorus, %	0.57	0.65	0.65	0.61	0.57	0.62	0.62	0.60	0.62	0.63	0.63	0.63	-
Available phosphorus, %	0.44	0.44	0.44	0.44	0.40	0.40	0.40	0.40	0.42	0.42	0.42	0.42	0.43
Lysine dig, %	1.44	1.44	1.44	1.44	1.33	1.33	1.33	1.33	1.30	1.30	1.30	1.30	1.21
Methionine (Met) dig, %	0.52	0.60	0.60	0.56	0.49	0.54	0.54	0.52	0.51	0.53	0.53	0.52	-
Met+ cysteine dig, %	0.85	0.85	0.85	0.85	0.47	0.52	0.52	0.49	0.48	0.50	0.50	0.49	0.69
Threonine dig, %	0.94	0.94	0.94	0.94	0.79	0.79	0.79	0.79	0.77	0.77	0.77	0.77	0.78
Tryptophan dig, %	0.25	0.25	0.25	0.25	0.85	0.85	0.85	0.85	0.84	0.84	0.84	0.84	0.23
Valine dig, %	0.99	0.99	0.99	0.99	0.24	0.24	0.24	0.24	0.25	0.25	0.25	0.25	0.83
Sodium, %	0.42	0.31	0.31	0.37	0.89	0.89	0.89	0.89	0.81	0.81	0.81	0.81	0.25
Zinc, %	0.25	0.25	0.25	0.25	0.34	0.28	0.28	0.31	0.30	0.30	0.30	0.30	-
Copper, %	0.02	0.02	0.02	0.02	0.25	0.25	0.25	0.25	3.35	3.35	3.35	3.35	-

^1^ Levels per kg of Vitamin Premix: vitamin A (min) 6000 IU; vitamin D_3_ (min) 1500 IU; vitamin E (min) 15,000 mg; vitamin K_3_ (min) 1500 mg; vitamin B1 (min) 1350 mg; vitamin B2 4. 000 mg; vitamin B6 2000 mg; vitamin B12 (min) 20 mg; niacin (min) 20,000 mg; pantothenic acid (min) 9350 mg; folic acid (min) 600 mg; biotin (min) 80 mg; selenium (min) 300 mg; ^2^ levels per kg of Mineral Premix: iron (min) 100 mg; copper (min) 10 mg; manganese (min) 40 g; cobalt (min) 1000 mg; zinc (min) 100 mg; iodine (min) 1500 mg; ^3^ milk powder (Auster, Hortolândia, Brazil); ^4^ antioxidant (Alltech, Maringá, Brazil).

**Table 2 animals-14-00631-t002:** Performance metrics for piglets fed different diets: live weight, daily weight gain, daily feed intake, and feed conversion ratio.

Parameters	Treatments				CV (%)	*p*-Value
	BP	AY	AYI	BPAY		
Pre-Starter I (21–28 days)						
LW21d (kg)	5.246	5.248	5.243	5.249	15.6	0.9664
DWG (kg/day)	0.165	0.127	0.141	0.165	26.6	0.1123
DFI (kg/day)	0.228 ^a^	0.174 ^b^	0.188 ^b^	0.219 ^ab^	24.4	0.0854
FCR	1.342	1.320	1.341	1.333	7.9	0.7458
Pre-Starter II (29–35 days)						
LW29d (kg)	6.404	6.139	6.233	6.402	14.9	0.9054
DWG (kg/day)	0.278	0.291	0.290	0.289	12.3	0.8125
DFI (kg/day)	0.423	0.396	0.398	0.417	15.0	0.8456
FCR	1.484 ^a^	1.358 ^b^	1.379 ^ab^	1.442 ^ab^	8.7	0.0152
Starter I (36–42 days)						
LW36d (kg)	8.352	8.178	8.265	8.423	13.5	0.9857
DWG (kg/day)	0.387	0.374	0.402	0.378	12.6	0.4189
DFI (kg/day)	0.648	0.600	0.629	0.635	12.1	0.78247
FCR	1.617	1.606	1.574	1.692	9.3	0.1645
Pre-Starter I e II, Starter I (21–42 days)						
DWG (kg/day)	0.284	0.264	0.278	0.277	12.2	0.9412
DFI (kg/day)	0.433	0.390	0.405	0.424	13.8	0.4987
FCR	1.529 ^b^	1.475 ^a^	1.458 ^a^	1.528 ^ab^	5.9	0.0540
Starter II (42–63 days)						
LW43d (kg)	11.204	10.793	11.078	11.068	12.5	0.9147
DWG (kg/day)	0.570	0.560	0.569	0.584	12.2	0.9206
DFI (kg/day)	0.907	0.880	0.905	0.908	11.7	0.9546
FCR	1.594	1.575	1.575	1.556	3.4	0.5236
LW63d (kg)	23.238	22.607	23.028	23.327	11.7	0.9036
Total (21–63 days)						
DWG (kg/day)	0.427	0.412	0.423	0.430	11.3	0.8547
DFI (kg/day)	0.670	0.635	0.655	0.666	11.8	0.8812
FCR	1.571	1.541	1.549	1.546	3.2	0.1250

^a^, ^b^ Distinct letters in rows indicate significant differences (*p* ≤ 0.05) and trends (*p* ≤ 0.10) as determined using the General Linear Model (GLM) ANOVA and Tukey’s post-test for all data. CV = coefficient of variation. Daily weight gain (DWG); daily feed intake (DFI), live weight (LW), feed conversion (FCR). Blood plasma (BP), autolyzed yeast (AY), autolyzed yeast plus immunomodulator (AYI), blood plasma and autolyzed yeast (BPAY).

**Table 3 animals-14-00631-t003:** Mean values for pro-inflammatory and anti-inflammatory cytokines and intestinal permeability of piglets fed blood plasma (BP), autolyzed yeast (AY), autolyzed yeast plus immunomodulator (AYI), and blood plasma and autolyzed yeast (BPAY).

	Treatments				CV (%)	*p*-Value
	BP	AY	AYI	BPAY		
IFN-α (pg/mL)	1.243 ^ab^	3.124 ^a^	1.234 ^b^	1.672 ^ab^	137.1	0.0612
IFN-γ (pg/mL)	2.478	2.478	2.478	2.478	0.0	1.0000
IL-10 (pg/mL)	2.976 ^b^	6.453 ^ab^	4.212 ^ab^	8.280 ^a^	92.8	0.0723
IL-1β (pg/mL)	4.638	5.466	6.841	2.802	154.3	0.7521
IL-4 (pg/mL)	1.631	1.902	1.903	0.786	137.3	0.2014
IL-6 (pg/mL)	24.713	81.630	49.296	22.498	190.7	0.9523
IL-8 (pg/mL)	86.868	104.739	86.601	129.692	71.2	0.2874
TNF-α (pg/mL)	6.741	14.843	3.434	13.386	164.3	0.1289
IL-12p40 (pg/mL)	450.860	466.762	573.447	499.656	51.7	0.4920
FITC-dextran (μg/mL)	0.3980	0.3646	0.3731	0.3840	20.5	0.3741

^a^, ^b^ Determined using the General Linear Model (GLM) ANOVA and Tukey’s post-test (*p* ≤ 0.10). CV = coefficient of variation.

**Table 4 animals-14-00631-t004:** Diarrhea and thinness indices and scores, number of animals medicated, and medications used in piglets fed blood plasma (BP), autolyzed yeast (AY), and autolyzed yeast plus immunomodulator (AYI), and blood plasma and autolyzed yeast (BPAY) from 21 to 63 days of age.

	Treatments				*p*-Value
	BP	AY	AYI	BPAY	
	Diarrhea				
Score 2 (n)	2	3	1	3	0.5600
Score 3 (n)	9	9	2	8	0.1401
Score 2 + 3 (n)	11 ^a^	12 ^a^	3 ^b^	11 ^a^	0.0816
Diarrhea index	0.183	0.200	0.050	0.183	-
	Thinness				
Score 1 (n)	2	4	5	5	0.6593
Score 2 (n)	2	1	2	0	0.5246
Score 1 + 2 (n)	4	5	7	5	0.8043
Thinness index	0.066	0.083	0.116	0.083	-
	Medicines				
Medicated (n)	7	2	3	2	0.1621
Medications (n)	17 ^a^	3 ^c^	8 ^b^	2 ^c^	0.0001

^a, b, c^ Distinct letters in the rows indicate difference according to Chi-square test (*p* ≤ 0.05).

**Table 5 animals-14-00631-t005:** Mean values of blood parameters, i.e., glucose, triglycerides, cholesterol, insulin, leptin, beta hydroxybutyrate (BHB) and non-esterified fatty acids (NEFAs) at 35 and 49 days of age of piglets fed blood plasma (BP), autolyzed yeast (AY), autolyzed yeast plus immunomodulator (AYI), and blood plasma and autolyzed yeast (BPAY).

Parameters	Treatments				CV (%)	*p*-Value
	BP	AY	AYI	BPAY		
	1st collection (35 days old)					
Glucose mg/dL	127.67	142.25	123.25	132.42	25.9	0.8804
Triglycerides mg/dL	41.17 ^b^	73.83 ^a^	69.92 ^ab^	69.58 ^ab^	56.0	0.0400
Cholesterol mg/dL	85.33	81.00	75.33	86.75	26.6	0.6467
Insulin pmol/L	28.58	35.92	23.54	18.10	67.2	0.1692
Leptin pg/mL	3976.98	2752.54	1800.32	2078.72	151.5	0.4530
NEFA	0.68	0.44	0.44	0.39	79.6	0.5424
BHB ng/μL	117.50 ^a^	79.96 ^b^	103.36 ^ab^	84.67 ^b^	32.0	0.0072
	2nd collection (49 days old)					
Glucose mg/dL	103.82 ^b^	112.52 ^ab^	142.41 ^a^	131.89 ^ab^	27.5	0.0878
Triglycerides mg/dL	65.19	86.64	77.15	79.56	40.2	0.4760
Cholesterol mg/dL	112.01	101.07	94.66	108.95	32.4	0.5894
Insulin pmol/L	69.32	75.78	87.21	62.64	59.5	0.8603
Leptin pg/mL	203.42	209.60	90.83	124.76	112.6	0.3254
NEFA	0.65	0.62	0.72	0.53	51.7	0.8120
BHB ng/μL	34.70	39.18	26.14	40.50	68.3	0.5187

^a^, ^b^ Determined by using the General Linear Model (GLM) ANOVA and Tukey’s post-test (*p* ≤ 0.05) and trend (*p* ≤ 0.10). CV = coefficient of variation.

## Data Availability

The datasets generated during and/or analyzed during the current study are available from the corresponding author on reasonable request.

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
