# Peer review of "Enhancing Weaned Piglet Health and Performance: The Role of Autolyzed Yeast (Saccharomyces cerevisiae) and β-Glucans as a Blood Plasma Alternative in Diets"

_animals, 2024, doi:10.3390/ani14040631_

Round 1
Reviewer 1 Report
Comments and Suggestions for Authors
The research is aimed at studying the Blood plasma replacement through autolysed yeast (Saccharomyces cerevisiae) in the diet of weaned pigs: effects on performance, intestinal health and metabolism.
However, some remarks should be made.
1. Title of the manuscript. Should be fixed. The name is misleading about the use of blood plasma.
2. In the section materials and research methods there is no description of the blood plasma used. What are its characteristics, including nutritional value!
3. Revise the conclusion. It does not analyze the results obtained with specific meanings. As well as assessing the prospects for further application of the results obtained in practice. How realistic is it to apply the results obtained in practice?
4. How the physiological and biochemical parameters of the body of experimental animals will change in the future. Do you mean after using investigational drugs? Will this lead to any changes?
Comments on the Quality of English Language
The manuscript is written in good and accessible English.
Special and professional terms and phrases are widely used in the text.
However, minor text checking is required.
Author Response
Dear Reviewer
Thank you for your insightful suggestions and corrections, which have been instrumental in enhancing the clarity and quality of our manuscript.

Reviewer 2 Report
Comments and Suggestions for Authors
This paper is well written and contributes to the knowlegde of the nutrition of the early weaned piglet. Also an extensive discussion showed carefull interpretation of the results . However I would have liked more integration of the different aspects.
Although the authors in there discussion explain that the results are in line with the literature, I feel that this experiment should be repeated to be really conclusive.
I would have preferred a factorial experimental design in which the 2 factors of autolyzed yeast and the immunomodulator and their interaction were studied.
I feel that the title should be altered. The blood plasma was not only replaced by AY alone but also by soyprotein, pure amino acids and dical.
Since it is a paper on feed ingredients the authors should provide more information about the quality of the ingredients.. What are the dig amino acid contents of AY and BP? They may vary as was referenced in the paper by Almeida e.a.(15). The quality (digestibility) of the BP may affect the results for that treatment.
I have doubts regarding the contribution of the all data presented in table 2.
The starting weight at day 21 are almost equal. That is good. The fact that the data are presented by weeks causes a large variation. In the period 21-28 days the htere were trends for differences in feed intake and thus DWG. This affects also the FCR in the period 29-35. therefore the data from day 28 should be used as covariable.
I feel that the data 21-42 are the most valuable.
Treatment BP had the most medication and also other parameters indicate that this teatment had the most problems. I feel that the metabolic differences ( triglycerides and BHB) could be explained by the lower fat digestibility and therefore causing less triglycerides and more BHB in blood.
Minor remarks:
the page numbers are not consistent
Is reference 26 correct? are Suinos and Aves changed?
Author Response
Dear Reviewer
Thank you for your valuable comments and suggestions regarding our manuscript. We acknowledge the importance of integrating different aspects of our study and appreciate your perspective on the necessity for further research. In response to your concerns, it's pertinent to highlight that our experiment was conducted in a research facility designed to replicate commercial swine production conditions. This approach was essential to ensure that our findings are applicable and relevant to real-world scenarios in swine nutrition and management.
Regarding the experimental design, we strategically chose a sample size (n=12) that is statistically representative, aiming to avoid the common pitfalls associated with small sample sizes. This careful planning was a step in ensuring the validity and reliability of our results, thereby mitigating potential errors, and enhancing the robustness of our conclusions.
Moving forward, we recognize the value of additional research in this area. Our team is in the process of designing future studies, which will explore varying levels of the nutritional ingredients in question. These studies are intended to further substantiate our current findings and to broaden the scope of our understanding in this field.
We are grateful for your insights, which have been instrumental in refining our approach and will undoubtedly contribute to the advancement of knowledge in swine nutrition research.
Best regards

Reviewer 3 Report
Comments and Suggestions for Authors
Authors evaluated the effect of blood plasma replacement through autolysed yeast in weaned piglets. The argument of the study is very interesting, and it provides new insights related to the effect of autolysed yeasts with or without β-glucans on animal performance, health and gut microbiota. Feed additives alternative to antibiotics are an important field that require more data and findings according to the following study. However, the paper could not be accepted in the present form, since different points needs a clarification prior to reconsider the manuscript for the publication. Please, find my comments point-by-point below.
Line 18: According to the title, did your article involved nursery or weaned piglets?
Line 29: Did you mean weaning?
Abstract: I have found an interesting solution to provide results using the percentage to express the improvement or the decrease in comparison to the control group (BP). However, I believe that in the abstract section will be more useful to provide the data with their original unit of measurement in order to be useful for readers for making comparisons with other studies.
Line 45: The p-value is missing.
Line 53: Add the space between segregation and the references [1,2].
Line 60-61: The sentence is a bit complicated, please rephrase.
Line 67: Avoid too many repetitions of the verb to increase.
Line 68: The post-weaning phase is characterized by a high occurrence of infectious diseased caused by different virus (Rotavirus) and pathogenic bacteria (E. coli) with manifestation of the typical post-weaning diarrhoea and possible systemic disorders such as oedema disease due to the presence of enterotoxigenic strains of E. coli. I believe that the aspects of pathogenic bacteria will be important to be mentioned in order to strengthen your research in ameliorating health status thus decreasing antibiotics use and antimicrobial resistance spreading.
Line 70: feed additives.
Line 81: In this phase IgGs cannot be adsorbed by the gut that already established its structure building its specific junctions. In addition, IgGs in plasma are not specific for the present pathogens at a farm level.
Line 84: Please specify the difference between probiotic and prebiotic, and its relation to S. cerevisiae.
Line 90: How are the autolysed yeasts obtained?
Line 92: “dry matter basis”.
Line 92: lysine and glutamate DM basis?
Line 98: Specify in the EU according to Reg 2003/1831 and recently there are more limitations on veterinary drugs according to Reg EU 2019/6.
Line 103: Can you provide a convincing reason for the novelty of the following study?
Line 113: Provide the name of the company.
Line 117: Which was the ratio 1,3 and 1,6 β-glucans?
Line 128: Why did not including BP+immunomodulator?
Line 132: Did animals were weaned at 22 days?
Line 142: Did you provide meal feed? Dry or wet feed?
Table 1: Provide the %CP of soybean meal. What is Star Pro and Banox? You tried to substitute the BP but your CP is still high in the used formula. Why do not reduce the total amount of CP for a better aminoacidic balance?
Line 155: According to the results I guess you used a faecal score scale. Please describe the used scale briefly.
Line 173: I suggest changing the title in intestinal permeability evaluation and blood sampling.
Line 175: 48 in total. Please provide the dosage. The name of the compound should be provided in extenso before the acronym. Describe after how much time did you titre the concentration of FITC-d in the blood serum.
Line 178: jugular vein.
Line 187: How did you define “deep rectum”?
Line 193: Provide the information of the company (e.g. Name, City, State).
Line 194: Did you evaluate the ratio 260/280?
Line 197: Provide primers sequence or the reference article.
Line 205: Do you have a reference for your pipeline?
Statistical analysis: Describe the evaluation of categorical data. Please the description of ANOVA is too simplistic, did you include the effect of time, treatment and their interaction? Did you include the effect of the phase?
Line 235: Remove the not significant p-values (check it in all the text).
Line 241: Avoid using “better”, I suggest substituting this term with more appropriate “decreased”. This sentence should be revised for the English language.
Table 2: Please include a title of the table, the reported information are useful as footnotes, and they should be moved below the table. Instead of CV% it will be important to provide the standard error or the standard deviation. Correct the measurement units (DWG, kg/day; DFI kg/day). Data analysed for median differences (Kruskal Wallis) should not be reported as means. How did you obtain a so high efficiency during the first phase (FCR < 1.5)?
Line 256: Are medications intended as antibiotic treatments?
Line 259: In which period these differences were detected?
Table 3: Data analysed for median differences (Kruskal Wallis) should not be reported as means.
Table 4: It is important to define the period related to these data. Why did you detect similar diarrhoea frequencies in BP and AY but you treat animals differently (17 vs 3).
Figure 2: I have found confounding the use of different labelling for treatment groups. Please modify the figure using BP, AY, AYI, and BPAY to be consistent.
Line 316-320: Refereed to which period? Did you consider the entire trial or a specific timepoint? Did you sample faeces only after 14 days? Why do not wait until 42 days? Different studies have shown that a period of three four weeks could be considered favourable for detecting modifications of the gut microbiota.
Discussion: The rationale of the selected concentrations used for AY and immunomodulator should be clarified and the reason behind changing their concentrations in each phase.
Line 419: Could you provide the price comparison for AY and BP?
Line 424: Since the introduction and the aim of the study was focused also on the evaluation of immunomodulator (β-glucans), why do not titre the concentration of IgG?
Line 438: Did you conduct the trial in an experimental centre? Can the low concentration of cytokines be referred to high-welfare conditions of animals (e.g. environmental enrichments, correct handling, air quality, space…)
Line 455: According to your data, BP given the same results of AY in terms of diarrhoea.
Line 467: Which kind of glycoprotein?
Line 469: FITC-d did not highlight different intestinal permeability among groups.
Lines 474-479: How do you explain the difference with your study?
Line 480: Please define the drug used in your study (antibiotics, NSAID).
Line 486: Why do not measure the antioxidant barrier status of plasma and IgG concentration?
Line 507: Did you consider the detected increase in triglycerides an indicator of positive health status?
Line 520: Is there a correlation between higher triglycerides and lower BHB?
Line 594: Avoid using “relevant” and provide a clear statement regarding the most important findings of your study. Based on your expertise, which concentration or which combination do you consider that provided the best results in your trial? Would you suggest using the combination of AY and immunomodulators as valuable feed additive at a farm level or only the AY could provide enough benefits for animal health and performance?
Comments on the Quality of English Language
Overal quality of writing is good, there are only few sentences that required to be revised/rephrased for improving the quality of the English.
Author Response
Dear reviewer
Thank you for your thoughtful and constructive comments. We greatly appreciate the time you took to evaluate our manuscript and provide detailed feedback. Your insights have been very important in guiding our revisions, significantly enhancing the overall quality of our work. We recognize the importance of your suggestions and have diligently worked to address each point raised to ensure our paper meets the high standards required for publication. Your input has not only improved the clarity and depth of our study but also contributed to the broader understanding of feed additives as alternatives to antibiotics in animal nutrition.
Best regards

Round 2
Reviewer 3 Report
Comments and Suggestions for Authors
The authors revised the manuscript according to my comments, and I have sincerely appreciated their effort to improve the quality of the paper.
I have only few comments prior to reconsider the manuscript ready for the publication.
In the abstract section, the p-values for statistically significant differences should be added. My previous comment was indicating that, if absent, the p-values should have been added.
Lines 87-89: This statement is correct, but it should benefit from a reference.
AUT# In response to your query regarding the novelty of our research, we have expanded our analyses to include contemporary and sophisticated evaluations of intestinal health, such as microbiome analysis, immune response (cytokine profiling), intestinal absorption, and clinical signs associated with the dietary ingredients. We believe that it is of significant scientific interest to increase the body of research that illuminates the aspect of intestinal health in piglets. Our study contributes to this field by providing detailed insights into how dietary interventions can influence gut health, potentially leading to improved animal welfare and production efficiencies. This approach not only addresses a critical gap in current research but also aligns with the growing emphasis on sustainable and health-conscious livestock management practices.
Reviewer: Thank you for your kind answer. I believe that the novelty of the research could be briefly resumed at the end of the introduction section:
Table 1: Please provide the description of Banox. If it is a commercial product, please provide the company name and location.
Line 250: “Dunn’s post-hoc test”.
Table 2: I really appreciated your response. However, I am still convinced that, even if redundant, the correct unit of measurement is kg/day for DWG and DFI and not only kg as LW.
AUT# Yes, the term 'medications' in this context refers to treatments such as antibiotics and anti-inflammatories, therapeutic use.
Reviewer: Thank you for the answer, please state in the text that medications is not intended only for antibiotic treatments.
Figure 1: The labelling of figure 1 could be modified similarly to the figure 2 with BP, AY, AYI and BPAY.
AUT# The fecal sampling period in our study was at 35 days of age, which corresponds to 14 days post-weaning. This specific timeframe was chosen as the primary aim was to evaluate the rapid recovery of piglets from the stress of early weaning, including the balancing of their gut microbiota, particularly by fostering beneficial bacterial populations. Therefore, fecal samples were collected exclusively 14 days after weaning. However, your point is well-taken. It is acknowledged that the intestinal microbiota continues to evolve with age and dietary composition, and assessing these changes at later stages could provide additional valuable insights.
Reviewer: Please underline that your results are referred only after the recovery period from early weaning, but other changes in the microbial community could have been observed at the end of the trial.
AUT# The primary focus of our study was to substitute blood plasma (BP) with autolyzed yeast (AY). Given the frequent use of BP as an ingredient at levels similar to those in our study, we opted to replace it with AY at equivalent levels. As for the immunomodulator dosage, we adhered to the commercial recommendations provided by Biorigin, the manufacturer of the product.
Reviewer: Thank you, this information should be included into the manuscript.
Author Response
Dear,
We would like to express our sincere appreciation for your constructive comments and suggestions on our manuscript. We have carefully addressed each of your comments and incorporated the necessary changes into the revised manuscript. To assist in your review, we have highlighted all modifications in green.
We are grateful for the opportunity to improve our manuscript based on your feedback and are confident that these changes align with your expectations.
We remain at your disposal for any further information or clarification that may be needed regarding our submission.
Thank you
